# RNAi-based screens uncover a potential new role for the orphan neuropeptide receptor Moody in *Drosophila* female germline stem cell maintenance

**Tianlu Ma, Shinya Matsuoka¤, Daniela Drummond-Barbosa**[ID]*

Department of Biochemistry and Molecular Biology, Bloomberg School of Public Health, Johns Hopkins University, Baltimore, Maryland, United States of America

¤ Current address: AstraZeneca K. K., Osaka, Japan

* dbarbosa@jhu.edu

**Data Availability Statement:** All relevant data are within the paper and its Supporting Information files.

## Abstract

Reproduction is highly sensitive to changes in physiology and the external environment. Neuropeptides are evolutionarily conserved signaling molecules that regulate multiple physiological processes. However, the potential reproductive roles of many neuropeptide signaling pathways remain underexplored. Here, we describe the results of RNAi-based screens in *Drosophila melanogaster* to identify neuropeptides/neuropeptide receptors with potential roles in oogenesis. The screen read-outs were either the number of eggs laid per female per day over time or fluorescence microscopy analysis of dissected ovaries. We found that the orphan neuropeptide receptor encoded by *moody* (homologous to mammalian melatonin receptors) is likely required in somatic cells for normal egg production and proper germline stem cell maintenance. However, the egg laying screens had low signal-to-noise ratio and did not lead to the identification of additional candidates. Thus, although egg count assays might be useful for large-scale screens to identify oogenesis regulators that result in dramatic changes in oogenesis, more labor-intensive microscopy-based screen are better applicable for identifying new physiological regulators of oogenesis with more subtle phenotypes.

## Introduction

Reproduction is highly responsive to changes in physiology and the external environment [1]. In mammals, many of these changes impinge on the hypothalamic-pituitary-gonadal axis, the central regulator of reproduction. Gonadotropin-releasing hormone (GnRH) is produced and secreted by neurosecretory cells in the hypothalamus and acts on the anterior pituitary gland to stimulate the release of follicle stimulating hormone (FSH) and luteinizing hormone (LH) [2]. Obesity and excessive exercise can both lead to reduced gonadotropin levels in humans [3], and psychological stress decreases LH and FSH levels in rodents and other mammals [4]. Neuropeptides are an important group of signaling molecules that lie at the intersection of the hypothalamic-pituitary-gonadal axis and physiology. For example, the neuropeptide

**Funding:** This work was supported by National Institutes of Health (NIH) grants R01 GM069875 (D.D.-B) and R01 GM125121 (D.D.-B.). T.M. was supported by NIH training grant T32 CA009110. The funders had no role in study design, data collection and analysis, decision to publish, or preparation of the manuscript.

**Competing interests:** The authors have declared that no competing interests exist.

kisspeptin, encoded by *KISS1*, is a key activator of GnRH secretion and is in turn regulated by additional neuropeptides such as neuropeptide Y (NPY) [5]. Kisspeptin is also regulated by systemic factors, including insulin and adipocyte-derived leptin [2]. Neuropeptides likely have more ancient roles in communicating physiological state from the brain to the gonad that are independent of the hypothalamic-pituitary-gonadal axis, considering that this axis is a more recent evolutionary addition to the physiology of reproduction.

Neuropeptides are evolutionarily conserved signaling molecules present from invertebrates to humans [6]. There are over 100 neuropeptides in humans [7], while the *Drosophila melanogaster* genome encodes about 50 neuropeptides, some of which have been implicated in development, behavior, and reproduction [8,9]. *Drosophila* is a powerful model system for studying how physiology and the environment impact oogenesis [9]. Each *Drosophila* ovary is composed of 16 to 20 individual units called ovarioles, where follicles develop through 14 recognizable stages, including stage 8, when vitellogenesis begins, and stage 14, when the mature oocyte has fully formed dorsal appendages (Fig 1A). Follicles are formed in the anterior germarium, which houses two to three germline stem cells (GSCs) within a niche composed primarily of cap cells (Fig 1B). GSCs divide to self-renew and give rise to cystoblasts, which undergo four synchronous rounds of incomplete division to form 16-cell cysts composed of 15 nurse cells and one oocyte. Sixteen-cell cysts are enveloped by somatic follicle cells to form a new follicle (or egg chamber) that buds off from the germarium [9]. Several neuropeptides are known to control female reproduction. Neural-derived insulin-like peptides (ILPs) promote follicle growth and follicle cell proliferation in response to a nutrient rich diet [10], and insulin signaling is also required for GSC proliferation and maintenance, early germline cyst survival, and vitellogenesis [10–12]. ILP7 is also involved in oviposition [13], while gut-derived neuropeptide F (NPF; the *Drosophila* ortholog for NPY) regulates female GSC proliferation in response to mating and sex peptide (SP) signaling [14,15]. Multiple neuropeptides also regulate courtship and mating behavior [8]. The roles of additional neuropeptides/neuropeptide receptors in regulating *Drosophila* oogenesis, however, remain largely underexplored.

Here, we describe RNAi-based screens for neuropeptides/neuropeptide receptors regulating oogenesis. Three of these screens used egg counts as a read out, while a fourth smaller screen involved ovary dissection and microscopy analysis of specific oogenesis processes. The orphan neuropeptide receptor encoded by *moody* was identified as a new factor likely promoting egg production and GSC maintenance in both an egg count-based screen and the dissection-based screen. However, the egg count-based screens did not have sufficient signal-to-noise ratio to reliably identify additional novel regulators. Our results suggest that while egg counts can be valuable for screening large numbers of genes for major effects in oogenesis, the analysis of dissected ovaries is a more useful approach for identifying genes with more subtle physiological roles in specific steps of oogenesis.

## Results

### Egg count-based screens for neuropeptide/neuropeptide receptors with roles in oogenesis

To identify novel neuropeptide signaling pathways that might regulate oogenesis, we performed three separate screens using egg counts as a read-out: pan-neuronal neuropeptide knockdown using *nSyb-Gal4* [16], ubiquitous somatic neuropeptide receptor knockdown using *tub-Gal4* in combination with the temperature-sensitive Gal4 inhibitor *Gal80$^{ts}$* (*tub$^{ts}$*) [17], and germline-specific neuropeptide receptor knockdown using the maternal triple driver *MTD*, which combines three germline drivers (*otu-Gal4::VP16, nos-Gal4::VP16, and Gal4-nos. NGT*) expressed in the germarium and throughout oogenesis [18]. Zero-to-two-day-old

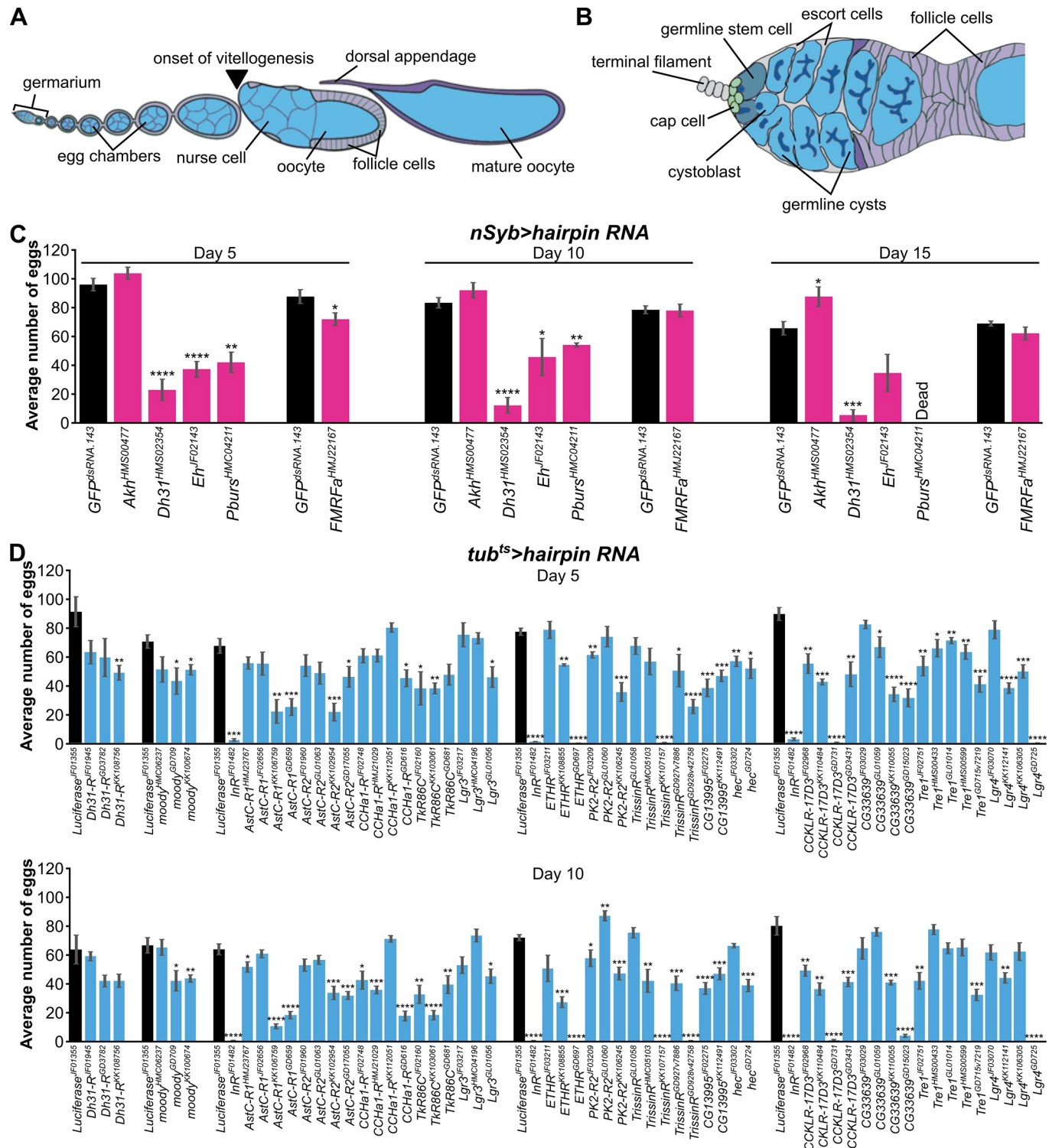

**Fig 1. RNAi-based screens to identify neuropeptides/neuropeptide receptors required for normal egg production.** (A) Diagram of *Drosophila* ovariole, which is composed of progressively more developed follicles (or egg chambers), ending with a mature stage 14 oocyte identifiable by its dorsal appendages. (B) Diagram of germarium, the anterior-most portion of the ovariole, which houses germline stem cells (GSCs) in a niche comprising of cap cells, terminal filament cells, and a subset of escort cells. GSCs divide to self-renew and give rise to cystoblasts, which undergo four synchronous divisions to form 16-cell cysts, each composed of 15 nurse cells and an oocyte. The 16-cell cysts become enveloped by follicle cells and bud off to form a new follicle. (C) Egg counts for pan-neuronal knockdown of the four candidate neuropeptide genes. *nSyb-Gal4* was used to drive *UAS-hairpin RNA* lines. The number of eggs laid per female per day were counted on days five, 10, and 15. *nSyb>GFP*^*dsRNA.143*^ served as control. (D) Egg counts for ubiquitous somatic knockdown of the 16 candidate

neuropeptide receptor genes. *tub-Gal80^ts; tub-Gal4* (*tub^ts*) was used to drive *UAS-hairpin RNA* lines. The number of eggs laid per female per day on days five and 10 are shown, with *tub^ts>Luc^JF01355* as control. *InR* knockdown served as an internal control. Day 15 data are included in S1 File. *$p<0.05$; **$p<0.01$; ***$p<0.001$; ****$p<0.0001$, Student's *t*-test. Data shown as mean±s.e.m.

females were paired with *y w* males at 29˚C on a nutrient-rich diet (wet yeast paste on molasses-agar plates), and the numbers of eggs laid in a 24-hour period on days five, 10, and 15 were counted. *nSyb>GFP^dsRNA.143* females (in which *nSyb-Gal4* drives expression of a *UAS-hairpin RNA* targeting *GFP* as a control RNAi) laid around 80.7±6.4 eggs at day five, consistent with the number of eggs laid by wild-type and control females on a rich diet [19,20], and were used as controls for the neuronal neuropeptide screen. By contrast, *tub^ts>GFP^dsRNA.143* females laid only 15.1±4.7 eggs at day five of control RNAi induction, much lower than previously reported data [19,20], prompting us to test additional control hairpin RNA lines with *tub^ts*. *tub^ts>Luc^JF01355* females (in which *tub^ts-Gal4* drives expression of a *UAS-hairpin RNA* targeting *Luciferase*, *Luc*, as a control RNAi) laid more eggs at five and 10 days after transgene induction than females with *tub^ts-Gal4* driving other control hairpin RNAs (S1A Fig) and were used as controls for the ubiquitous somatic neuropeptide receptor knockdown screen. We also tested multiple control *UAS-hairpin RNA* lines with *MTD-Gal4* and used *MTD>GFP^dsRNA.142* females, which laid the most eggs at day five (S1B Fig), as controls for the germline neuropeptide receptor screen.

We initially screened 36 neuropeptides and 46 total neuropeptide receptors (45 of which were screened in somatic cells and 20 in the germline) based on the design above. Knockdown of four neuropeptide receptors in the germline resulted in a statistically significant decrease in egg production on two different timepoints with at least one *UAS-hairpin RNA* line (S2 Fig and S1 File). Candidates from the initial neuropeptide neuronal knockdown or somatic neuropeptide receptor knockdown screens were tested a second time, and this secondary screen included additional genes that had not been initially screened (due to the late arrival of the corresponding fly stocks) (S1 File). Knockdown of four neuropeptides from the neuronal RNAi screen and 16 neuropeptide receptors from the somatic RNAi screen resulted in statistically significant decreases in egg production (Fig 1C and 1D and S1 File). Ubiquitous somatic *InR* knockdown resulted in nearly zero eggs laid and served as a positive control, and, consistent with previously published results [21], somatic knockdown of *ETHR* led to decreased egg production with two out of three independent RNAi lines. However, although the egg count assay allowed us to screen a total of 82 genes and over 100 *UAS-hairpin RNA* lines, there was not only a large variability in egg production among control lines (see above), but also among different *UAS-hairpin RNA* lines targeting the same gene. [For example, the number of eggs laid on day five with ubiquitous somatic knockdown of *torso* with four different *UAS-hairpin RNA* lines, all from the TRiP collection (fgr.hms.harvard.edu), ranged from 55% to 163% of the number of eggs laid by *Luc^JF01355* control.] These results indicated that the egg counting assay was excessively noisy and should ideally be validated with multiple *UAS-hairpin RNA* lines, genetic mutants, and additional ovarian analysis. Nevertheless, *Diuretic hormone 31* (*Dh31*) and *Diuretic hormone 31 Receptor* (*Dh31-R)*, which encode a ligand-receptor pair, emerged as candidates from the neuronal and somatic screens, respectively, leading us to focus next on these genes.

## Dh31/Dh31-R do not regulate *Drosophila* oogenesis

To directly examine the potential roles of *Dh31* and *Dh31-R* in oogenesis, we performed additional RNAi knockdown and genetic mutant analyses. We first determined knockdown efficiency of *UAS-Dh31-R hairpin* and *UAS-Dh31 hairpin* lines (Fig 2A and 2B) driven by *tub^ts* for

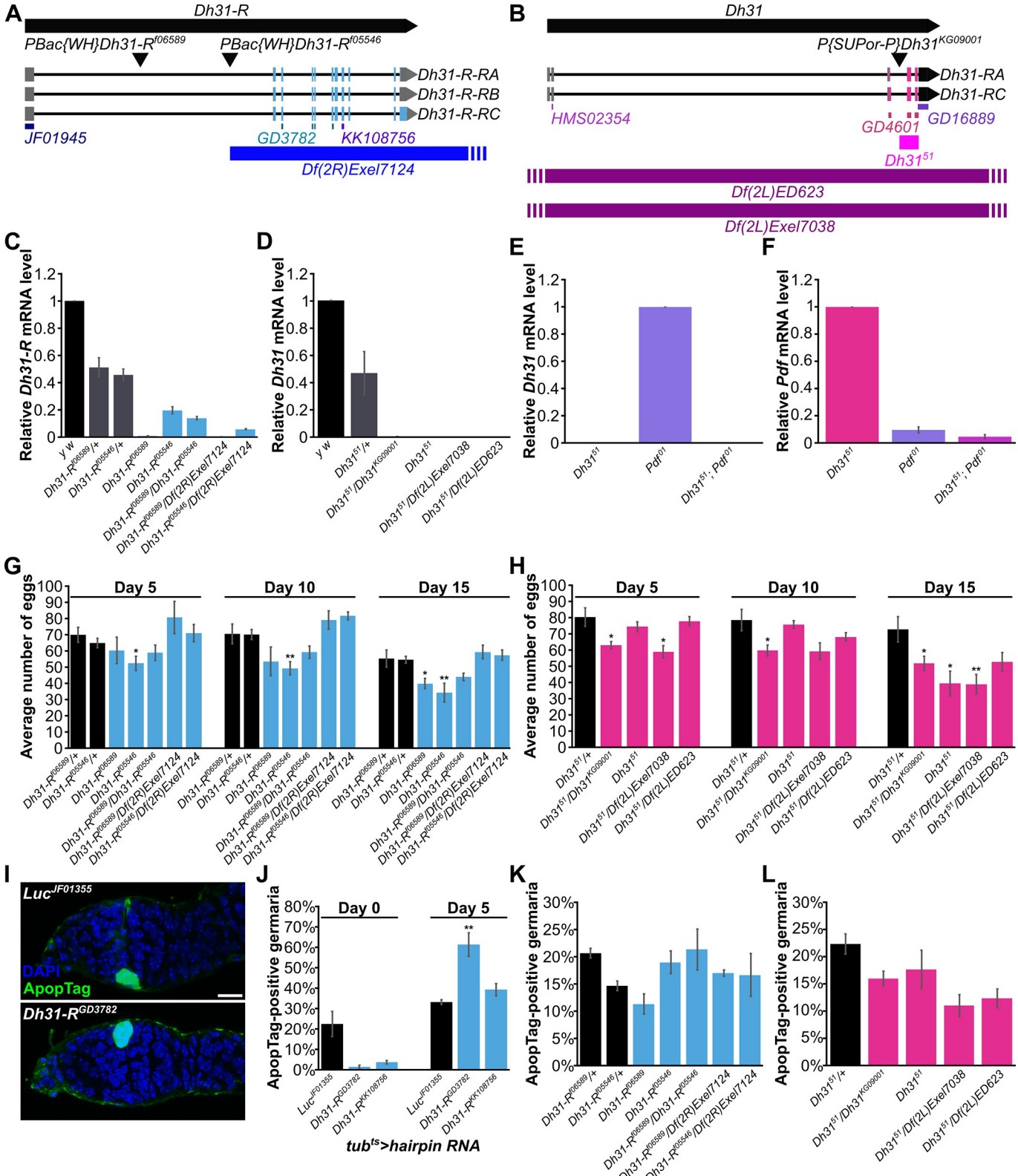

**Fig 2. _Dh31_ and _Dh31-R_ do not appear to regulate _Drosophila_ oogenesis.** (A) Schematic of the _Dh31-R_ gene, showing three mRNA isoforms for _Dh31-R_, which differ in the length of protein coding sequences in the last exon, with isoform RC having the longest protein-coding sequence and isoform RB having the shortest protein-coding sequence. The transposable element insertions _Dh31-R^{f05546}_ and _Dh31-R^{f06589}_ map to the 5' intronic region. The _UAS-Dh31-R^{JF01945}_

hairpin targets the common 5' UTR of *Dh31-R* transcripts, while *UAS-Dh31-R^GD3782* and *UAS-Dh31-R^KK108756* target protein-coding regions shared by *Dh31-R* transcripts. The *Df(2R)Exel7124* deficiency uncovers the entire protein-coding region of *Dh31-R*. (B) Schematic of the *Dh31* gene, showing two mRNA isoforms, which differ by three bases in the third exon. *Dh31^KG09001* is a strong hypomorphic allele resulting from a *P* element insertion in the third intron, and *Dh31^51* is a null allele created by imprecise excision of the *KG09001 P* element, resulting in a 735 bp deletion that removes most of the protein coding region. The *UAS-Dh31^HMS02354* hairpin targets the 5' UTR, *UAS-Dh31^GD4601* targets the coding region, and *UAS-Dh31^GD16889* targets the coding region and the 3' UTR. Deficiencies *Df(2L)ED623* and *Df(2L)Exel7038* uncover the entire *Dh31* gene region. (C-F) RT-qPCR analysis of *Dh31-R* (C), *Dh31* (D-E), and *Pdf* (F) transcript levels in seven-day-old female heads. Three biological replicates were analyzed for each genotype, with ten heads per biological replicate. (G,H) Egg counts for *Dh31-R* (G) and *Dh31* (H) mutants at 25˚C on days five, 10, and 15 after eclosion, showing no consistent differences between homozygous mutants and heterozygous controls. (I) Germarium at five days of knockdown by *tub^ts*-driven *Luc^JF01355* (top) or *Dh31-R^GD3782* (bottom), both showing a dying germline cyst labeled with ApopTag (green). Dying cysts found in *Luc^JF01355* control and *Dh31-R* knockdown germaria are visually similar based on ApopTag staining. DAPI (blue) labels nuclei. Scale bar, 10 μm. (J) Percentage of ApopTag-positive germaria in females with ubiquitous somatic knockdown of *Dh31-R* or *Luc* control at zero or five days of RNAi. (K-L) Percentage of ApopTag-positive germaria in five-day-old *Dh31-R* (K) and *Dh31* (L) mutant females compared to heterozygous controls at 25˚C. Three biological replicates were analyzed per genotype, with 100 germaria per biological replicate. $^*p<0.05$; $^{**}p<0.01$, Student's *t*-test. Data shown as mean±s.e.m.

seven days using RT-PCR. *Dh31-R^JF01945* did not induce knockdown of *Dh31-R*, but both *Dh31-R^GD3782* and *Dh31-R^KK108756* resulted in approximately 40% knockdown of *Dh31-R* (S3A Fig). *Dh31^HMS02354* resulted in an approximately 80% decrease in *Dh31* mRNA levels, the three *Dh31^GD4601* lines led to a 20–30% decrease, but neither of the two *Dh31^GD16889* lines led to a strong reduction in *Dh31* mRNA levels (S3B Fig). Ubiquitous somatic knockdown of *Dh31-R* led to a modest reduction in egg number compared to *Luc* knockdown control (S3C Fig), whereas neuronal knockdown of *Dh31* with all hairpin RNA constructs led to a statistically significant decrease in egg laying on day five (S3D Fig). However, the severity of the phenotype did not correlate with the level of knockdown observed. For example, although *Dh31^HMS02354* resulted in the strongest *Dh31* knockdown, *nSyb>Dh31^HMS02354* females laid more eggs than other females with less severe *Dh31* knockdown (S3B and S3D Fig). Thus, the RNAi analysis was inconclusive.

Owing to the inconsistencies and modest phenotypes observed in *Dh31* and *Dh31-R* knockdown females, we proceeded to analyze *Dh31* and *Dh31-R* genetic mutants (Fig 2A and 2B). Using RT-qPCR, we confirmed that *Dh31-R^f06589* and *Dh31-R^f05546* [22,23] are hypomorphic alleles (Fig 2C), and that *Dh31^51* [24] and *Dh31^KG09001* [25,26] are null and hypomorphic alleles, respectively (Fig 2D and 2E). Additionally, because published genetic evidence suggests that the neuropeptide PDF may also signal through Dh31-R [23], we tested *Dh31* and *Pdf* double mutants and confirmed that *Pdf^01* is a hypomorphic allele (Fig 2F) [27]. Using these validated genetic alleles, we did not observe any consistent decreases in egg production of *Dh31-R* or *Dh31* mutant females compared to heterozygous controls at 25˚C or 29˚C (the temperature at which previous RNAi experiments had been performed) (Fig 2G and 2H and S4 Fig). There were no significant differences in egg laying between *Dh31* and *Pdf* single and double mutants at 25˚C (S5A Fig); however, *Dh31^51; Pdf^01* double mutants lay significantly fewer eggs at 29˚C compared to either of the single mutants (S5B Fig). Dissection of *Dh31^51; Pdf^01* ovaries after five days at 29˚C revealed that the vast majority of ovarioles in some ovaries have more than one mature stage 14 egg, identified by their dorsal appendages (S5C Fig), suggesting that the decrease in egg laying is due to egg retention. This may indicate a potential role for *Dh31* and *Pdf* in ovulation.

In parallel to the egg count experiments above, we also performed some initial characterization of dissected ovaries to determine if specific steps of oogenesis might be disrupted by *Dh31-R* loss-of-function. While we did not notice any obvious changes in overall ovariole morphology, five days of ubiquitous somatic knockdown of *Dh31-R* using *Dh31-R^GD3782* (but not *Dh31-R^KK108756*) increased the numbers of dying early germline cysts (Fig 2I and 2J). Adding to these inconsistent results, analysis of *Dh31-R*, *Dh31*, and *Dh31^51; Pdf^01* mutants did not show any significant increase in the percent of germaria containing dying germline cysts at

either 25°C or 29°C compared to controls (Fig 2K and 2L and S6 Fig). Taken together, these data suggest that *Dh31* and *Dh31-R* do not regulate oogenesis.

### *moody* is likely required in somatic cells for proper maintenance of GSCs

We next focused on the orphan G protein coupled receptor (GPCR) *moody*. Ubiquitous somatic knockdown of *moody* with two out of three different hairpin RNA lines resulted in decreased egg production on days five, 10, and 15 of RNAi induction (Fig 1D and S1 File). Importantly, *moody* was identified in a separate RNAi-based screen aimed at identifying GPCRs with roles in regulating GSCs. In this screen, we tested a small subset of GPCRs that included five neuropeptide receptors (Table 1). We used either the ubiquitous somatic *tub*$^{ts}$ or the germline driver *nanos-Gal4::VP16* (referred to as *nos-Gal4*) [28] to drive *UAS-hairpin RNAs* against individual GPCRs and analyzed dissected ovaries for GSC loss or proliferation phenotypes at 10 days of RNAi knockdown. We identified *AstC-R1* and *moody* as potential regulators of GSC proliferation, based on the increased frequency of EdU-positive GSCs in females with germline-specific *AstC-R1* knockdown or ubiquitous somatic *moody* knockdown relative to controls (Table 1). However, these results were not reproducible for either *AstC-R1* or *moody* when EdU incorporation experiments were repeated using larger sample sizes and additional *UAS-hairpin RNA* lines with *MTD-Gal4* or *tub*$^{ts}$, respectively (S7 Fig). Nevertheless,

**Table 1. Results from RNAi-based screen for candidate GPCRs regulating GSC number and/or proliferation at 10 days of RNAi.**

| GPCR dsRNA* | Somatic knockdown (*tub*$^{ts}$) | | Germline knockdown (*nos-Gal4*) | |
|---|---|---|---|---|
| | GSC number | GSC proliferation | GSC number | GSC proliferation |
| **AkhR**$^{HMC03228}$ | No change | No change | No change | No change |
| **AkhR**$^{JF03256}$ | No change | No change | N.D. † | N.D. |
| **AstC-R1**$^{HMJ23767}$ | No change | No change | No change | Increase |
| **AstC-R1**$^{JF02656}$ | Decrease | No change | N.D. | N.D. |
| **CCHa1-R**$^{HMJ21029}$ | No change | No change | No change | No change |
| **CCHa1-R**$^{JF02748}$ | No change | No change | N.D. | N.D. |
| Dop1R1$^{HM04077}$ | No change | No change | N.D. | N.D. |
| Dop1R1$^{HMC02344}$ | No change | No change | No change | No change |
| Dop1R1$^{HMC05220}$ | No change | No change | No change | No change |
| Dop2R$^{HMC02988}$ | No change | No change | No change | No change |
| Dop2R$^{JF02025}$ | No change | No change | N.D. | N.D. |
| mAChR-C$^{GD717}$ | No change | No change | N.D. | N.D. |
| mAChR-C$^{HMJ23139}$ | No change | No change | No change | Decrease |
| mAChR-C$^{JF03291}$ | No change | No change | N.D. | N.D. |
| mGluR$^{HMS00191}$ | No change | No change | No change | No change |
| mGluR$^{HMS02201}$ | No change | No change | No change | No change |
| mGluR$^{JF01958}$ | No change | No change | N.D. | N.D. |
| **moody**$^{GD709}$ | Decrease | Increase | N.D. | N.D. |
| **moody**$^{KK100674}$ | Decrease | Increase | N.D. | N.D. |
| **moody**$^{GL01050}$ | N.D. | N.D. | No change | No change |
| mtt$^{HMS00367}$ | No change | No change | No change | Decrease |
| mtt$^{HMS02793}$ | N.D. | N.D. | Decrease | N.D. |
| **TkR86C**$^{GD681}$ | Decrease | Increase | N.D. | N.D. |
| **TkR86C**$^{JF02160}$ | No change | No change | N.D. | N.D. |

* GPCRs predicted to act as neuropeptide receptors are indicated in boldface.

† N.D., not determined.

the screen also showed that ubiquitous somatic *moody* knockdown using two different *UAS-moody hairpin* lines resulted in dramatic increase in GSC loss compared to control RNAi (Table 1), suggesting a potential role for *moody* in GSC maintenance.

*moody* encodes an orphan neuropeptide receptor (homolog of human melatonin receptor 1A and 1B, MTNR1A and MTNR1B [29]) required for integrity of the blood-brain barrier [30,31] and with no known roles in oogenesis. To follow up on the screen results suggesting a role for *moody* in GSC maintenance, we first performed RT-qPCR (using RNA from whole flies) to measure knockdown efficiency of the three *UAS-moody hairpin* lines expressed in the soma, which target distinct regions of *moody* (Fig 3A). Ubiquitous knockdown of *moody* driven by $tub^{ts}$ resulted in a 50–60% decrease in *moody* mRNA levels compared to $Luc^{JF01355}$ control (Fig 3B). *KK UAS-hairpin RNA* lines were created by site-specific insertion into the unannotated 30B site; however, a significant number of KK lines have an additional insertion at 40D, which causes ectopic expression of the transcription factor Tiptop and can lead to secondary effects [32,33], including loss of GSCs [34]. Therefore, we confirmed that the $moody^{KK100674}$ line has only the 30B insertion, and not the 40D insertion (Fig 3C), using previously described PCR methods [32].

Using these validated RNAi lines, we showed that ubiquitous somatic *moody* knockdown using $UAS\text{-}moody^{GD709}$ and $UAS\text{-}moody^{KK100674}$, but not $UAS\text{-}moody^{HMC06237}$, dramatically increased the rate of GSC loss compared to $Luc^{JF01355}$ control (Fig 3D–3F). Incidentally, ubiquitous somatic *moody* knockdown using $UAS\text{-}moody^{GD709}$ and $UAS\text{-}moody^{KK100674}$ (but not $UAS\text{-}moody^{HMC06237}$) also decreased egg production (Fig 1D). It is possible that $UAS\text{-}moody^{HMC06237}$ driven by $tub^{ts}$ does not cause GSC loss due to the presence of a *moody* suppressor in the genetic background. Alternatively, the $UAS\text{-}moody^{HMC06237}$ insertion might not be well expressed (to induce efficient knockdown of *moody*) in the specific tissue where *moody* is required for GSC maintenance. (See Materials and Methods for description of insertion sites of *UAS-moody hairpin* lines.) In fact, there is clear evidence that a given *UAS* transgene can be expressed at relatively different levels in distinct tissues depending on insertion site (e.g. [35]). On the other hand, it is unlikely that the GSC loss resulting from ubiquitous expression of $UAS\text{-}moody^{GD709}$ or $UAS\text{-}moody^{KK100674}$ represents off-target effects, as these hairpin lines target different regions of *moody* (Fig 3A). Moreover, the GSC loss phenotype was entirely dependent on induction of these hairpin lines by *tub-Gal4* because no GSC loss occurred when we analyzed *UAS-alone* controls (Fig 3G), further supporting the conclusion that *moody* is likely required somatically for GSC maintenance.

We next asked in what somatic tissue *moody* might be required for GSC maintenance. Because *moody* controls the blood-brain barrier [36] and insulin signaling controls GSC maintenance (through effects on cap cell numbers [11]), we first wondered if misregulation of insulin-like peptides produced in the brain might underlie the somatic *moody* GSC loss phenotype. However, we found no difference in cap cell numbers between *moody* and *Luc* control knockdown females (Fig 3H), indicating that somatic *moody* controls GSC numbers independently of changes in niche size (which is controlled by insulin signaling [11]), and thus ruling out effects on insulin signaling as a relevant mechanism. *moody* is widely expressed in adult *Drosophila* females, including at low levels in the ovary [37]. Previous single cell sequencing analysis of the adult ovary has detected *moody* expression in stretched cells (which cover the nurse cells in later follicles [38]) and corpus luteum cells [39], although expression in other ovarian populations of cells (e.g. niche) remained possible. Although somatic *moody* knockdown causes GSC loss in the absence of changes in cap cell numbers, it is conceivable that *moody* acts in cap cells or other nearby somatic cells to regulate GSC numbers. To determine if *moody* is required in neighboring somatic cells to regulate GSC numbers, we knocked down *moody* or *Luc* control using the niche driver *hh-Gal4* [40], which is expressed in terminal filament

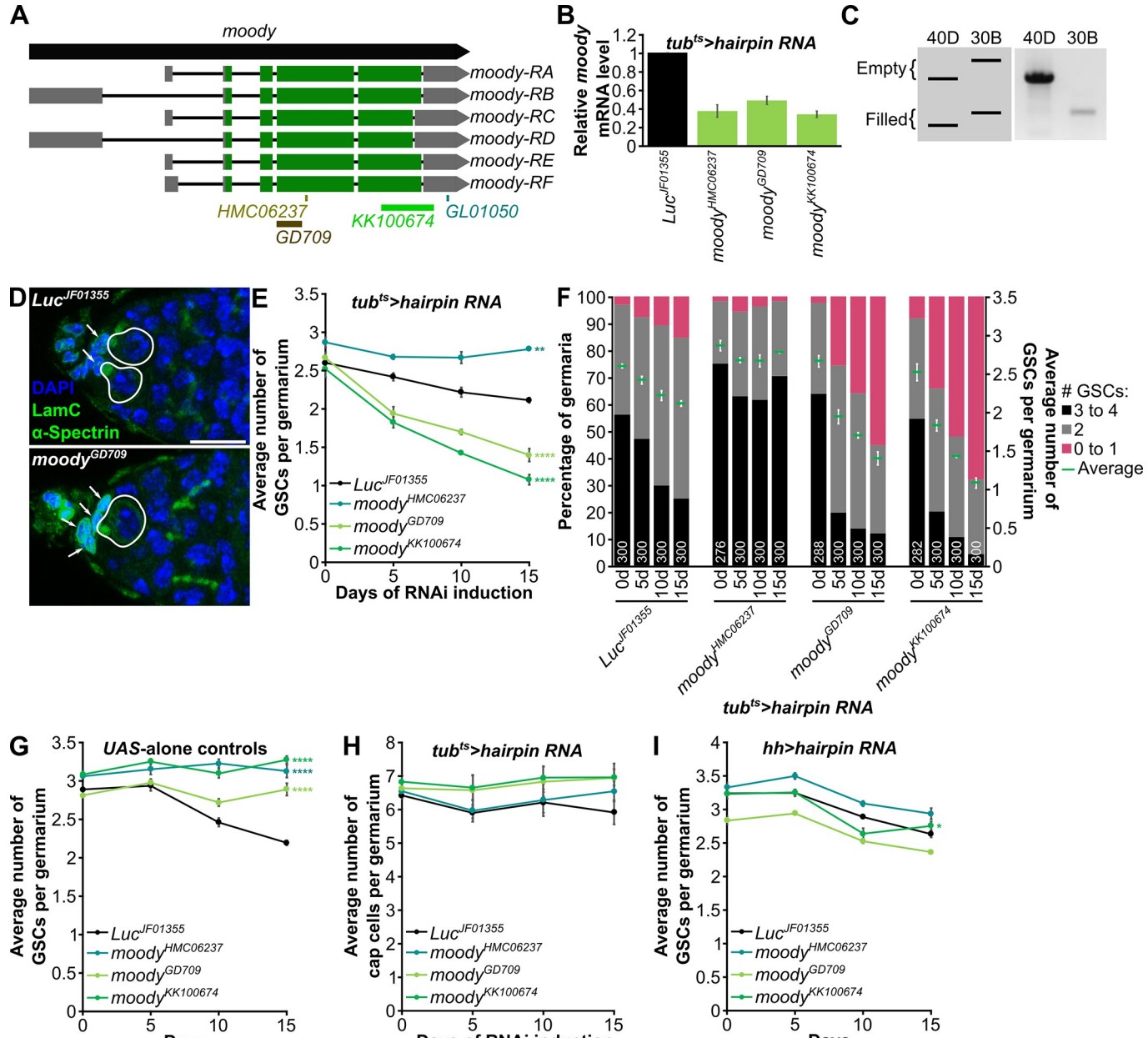

**Fig 3. *moody* appears to function in somatic cells to regulate GSC maintenance.** (A) Schematic showing *moody* gene and its six mRNA isoforms. The hairpin lines *UAS-moody^HMC06237* and *UAS-moody^GD709* target distinct portions of the protein coding regions shared by all transcripts, *UAS-moody^KK100674* targets part of the protein coding region and part of the 3' UTR, and *UAS-moody^GL01050* (optimized for germline expression) targets a portion of the 3' UTR sequences shared by all transcripts. (B) RT-qPCR analysis of *moody* from ovariectomized females at seven days of ubiquitous somatic *moody* or *Luc* control RNAi. Three biological replicates were analyzed, with 10 females per replicate. (C) Left: Schematic of expected PCR band sizes for KK lines. Right: PCR gel showing that *moody^KK100674* is inserted in the non-annotated 30B site, and not in the annotated 40D site. (D) Germaria from females at five days of ubiquitous somatic knockdown of *Luc* control or *moody* (using *moody^GD709*). DAPI (blue) labels nuclei. LamC (green), nuclear lamina of cap cells; α-Spectrin (green), fusome. Cap cells, arrows; GSCs, solid outlines. Scale bar, 10 μm. (E) Line graph showing the average number of GSCs per germarium at zero, five, 10, and 15 days of ubiquitous somatic knockdown of *moody* or control *Luc*. (F) Bar graph showing the percentage of germaria with zero to one, two, or more than three GSCs on the left *y*-axis and the average number of GSCs per germaria on the right *y*-axis at zero, five, 10, and 15 days of ubiquitous somatic knockdown of *moody* or control *Luc*. Same data are shown in (E and F). (G) Line graph showing the average number of GSCs per germarium at zero, five, 10, and 15 days after eclosion for *UAS*-alone controls of *moody* or *Luc* hairpin RNAi lines. (H) Line graph showing the average number of cap cells per germarium at zero, five, 10, and 15 days of ubiquitous somatic knockdown of *moody* or control *Luc*. (I) Line graph showing the average number of GSCs per germarium at zero, five, 10, and 15 days of *hh-Gal4*-driven knockdown of *moody* or control *Luc*. For (E-I), three biological replicates were analyzed for each genotype. The total number of germaria per genotype per timepoint for (E,F,H) are indicated in (F). For (G,I), a total of 300 germaria were analyzed per genotype per timepoint. *p<0.05; **p<0.01; ****p<0.0001, ANOVA: two-factor with replication. Data shown as mean±s.e.m.

cells, cap cells, and escort cells [35], and measured GSC numbers at zero, five, 10, and 15 days. However, we did not observe any differences in the rate of GSC loss with *moody* knockdown compared to *Luc* RNAi control (Fig 3I), indicating that *moody* is not required in somatic cells in the germarium to regulate GSC numbers. Given that the ovary is regulated by extensive inter-organ communication [9], it is conceivable that Moody might control GSC number by functioning in another tissue/organ through intermediate systemic modulators.

## Discussion

Neuropeptide signaling lies at the junction of reproduction and physiology [2,8,9]. In *Drosophila* females, insulin signaling has been shown to be critical for oogenesis [9], and SP/NPF signaling is responsible for increased GSC proliferation in response to mating [14,15]. Ecdysis-triggering hormone (ETH) persists in Inka cells in adults and regulates juvenile hormone (JH) synthesis and thus vitellogenesis [21]. Additionally, mating-induced changes in behavior are relayed by neuropeptide signaling [8]. For example, myoinhibiting peptide precursor (MIP) regulates female food preference upon mating [41], and enhanced Dh44 signaling delays sperm ejection in females [42]. In *Drosophila* males, corazonin (Crz) regulates ejaculation [43], and PDF/NPF signaling regulates mating duration in response to the presence of rival males [44]. In mammals, kisspeptin has emerged as a key regulator of GnRH production and secretion [45]. In addition to systemic signals such as leptin and insulin, multiple neuropeptides also regulate kisspeptin production itself [5]. Neurokinin B (NKB) activates kisspeptin-producing neurons [46], in addition to directly stimulating GnRH release [47]. NPY null mice have lower levels of *Kiss1* mRNA [48], while intracerebroventricular injection of NPY in male rats significantly increases *Kiss1* mRNA levels [49]. However, much remains to be learned about how this large family of signaling molecules and their receptors communicate physiological state and regulate reproduction. In this study, we found that the orphan neuropeptide receptor *moody* (which encodes the homolog of mammalian melatonin receptors) is likely required in the soma for GSC maintenance. Additionally, the results from our four screens show that while egg count assays are useful for screening large numbers of genes for severe oogenesis phenotypes, they do not adequately capture more subtle changes in oogenesis that can be detected through the detailed analysis of dissected ovaries.

### Screening for regulators of reproductive physiology can be challenging

Physiology relies on the convergence of multiple inputs/signaling pathways that coordinately regulate cellular processes/organ function throughout the body, including oogenesis [9]. Unlike the case for developmental phenotypes, where individual mutations often lead to severe blocks in development, genetic manipulations in a single physiological signaling pathway can often lead to relatively small phenotypic changes (e.g. in rates of certain oogenesis processes), which reflect its partial contribution in the context of a multitude of other integrated physiological inputs. It is therefore challenging to screen for genes regulating the physiology of oogenesis/reproduction. This study used two different screening strategies aimed at identifying potential physiological regulators of oogenesis: egg counting and ovary dissection analysis.

   Egg counting has been successfully used previously to identify regulators of fecundity and oogenesis. For example, egg count assays were used to show the negative effect of toxic chemicals such as cadmium and bisphenol A on egg production [50,51]. The effect of poor diet on egg production has also been well documented [19]. Egg counting has also been successfully used to identify amino acid transporters necessary in adipocytes and nuclear receptors required in the soma for egg production [17,20]. By contrast, our RNAi-based screens using egg counting as a readout indicate that this assay can have low signal-to-noise ratio and thus

be unreliable. In particular, egg production was very sensitive to genetic background, as different control *UAS-hairpin RNA* lines resulted in vastly different numbers of eggs laid (S1 Fig). The variability with genetic background is not surprising (given the large number of inputs that affect oogenesis) and has been previously reported by others [52]. The low signal-to-noise ratio makes the use of multiple *UAS-hairpin RNA* lines (to rule out off-target effects and ensure that phenotype penetrance correlates with knockdown efficiency) and other available genetic tools essential when using egg counts as a screening assay. Notably, while egg counts can be very useful for identifying genes or conditions that have large effects in oogenesis (e.g. *InR* and *ETHR*), our screens showed that the noise in the assay can overwhelm signals from more mild changes in oogenesis. In accordance, a previous study from our group found that ubiquitous somatic knockdown of *seven up* (*svp*, which encodes a nuclear receptor) led to decreased egg laying with effects in GSC maintenance, germline cyst survival, and vitellogenesis [17]. Interestingly, *svp* knockdown in adipocytes led to increased GSC loss and early germline cyst death, but no measurable effect on egg production, whereas *svp* knockdown in oenocytes caused degeneration of vitellogenic follicles and a reduction in number of eggs laid [17], indicating that egg count assays may not adequately capture more subtle changes in earlier steps of oogenesis. By contrast, we found that ubiquitous somatic knockdown of *moody* has a very strong effect on GSC maintenance (Fig 3D–3F) with consistent decreases in egg laying (Fig 1D). Similarly, insulin signaling in both the soma and germline are important for many processes during oogenesis, including vitellogenesis [9], and, consistent with those roles, knockdown of *InR* led to a significant reduction in egg laying in our screens (Fig 1D and S2 Fig). This again demonstrates that egg count assays can capture more dramatic ovarian phenotypes, although not necessarily more subtle phenotypes, which require analysis of dissected ovaries.

Dissection-based screens, despite their high resolving power, bring their own challenges. The fact that disruptions in specific signaling pathways might only have subtle effects on the rate of oogenesis processes demands larger sample sizes to identify these effects. While relatively large effects such as those seen with ubiquitous somatic *moody* knockdown on GSC maintenance were readily detectable even with one biological replicate, that replicate still comprised 80 germaria analyzed per genotype per timepoint. However, for phenotypes such as GSC proliferation (which relies on the frequencies of proliferation markers within the populations of GSCs analyzed), often hundreds of GSCs need to be analyzed in each experimental replicate to unambiguously determine whether the effect is indeed present. In fact, in this study, while both *AstC-R1* and *moody* seemed to regulate GSC proliferation based on our initial screen, we were unable to recapitulate the phenotype when using larger sample sizes, which included over 430 GSCs per genotype (S7 Fig). Thus, the large sample sizes needed to reliably screen for these phenotypes makes for very labor- and time-intensive screens. Despite these challenges, given how closely reproduction is tied to overall physiology, it remains vitally important to identify additional genes controlling the physiology of oogenesis. In particular, the importance of understanding how the brain sends physiological inputs to the ovary warrants further exploration of the complex roles of neuropeptide signaling in oogenesis. In future similar efforts to identify new ovarian regulators, dissection-based screens focusing on specific steps of oogenesis should ideally strike an optimal balance of labor and time invested in analyzing a moderate number of (rationally selected) candidate genes using sufficiently large sample sizes.

## Complex roles of neuropeptide signaling in whole-body physiology and reproduction

Neuropeptides are evolutionarily conserved signaling molecules that regulate a wide variety of physiological processes in organisms ranging from *C. elegans* to *Drosophila* to humans

[6,8,53]. In addition to a key role in regulating the hypothalamic-pituitary-gonadal axis and reproduction, mammalian neuropeptides also control circadian rhythm, water reabsorption, feeding behavior, stress, immunity, and even alcohol intake [54–58]. Activation of NPY signaling, for example, increases food intake and decreases stress and anxiety [54]. In *Drosophila*, NPF also regulates food intake, metabolism, and aggression, while other conserved neuropeptide orthologs, including Hugin (Neuromedin U homolog), SIFamide (gonadotropin inhibiting hormone GnIH homolog), and Drosulfakinin DSK (Cholecystokinin CCK homolog), regulate feeding behavior, taste and olfaction, learning and behavior, sleep, nociception, and alcohol tolerance [8]. However, only a handful of neuropeptide signaling pathways have been implicated in *Drosophila* oogenesis. Our egg count screens identified some potentially interesting candidates such as *AstC-R1*, *AstC-R2*, *CCHa1-R*, *TrissinR*, *CG13995*, *hec*, *PK2-R2*, *CCKLR-17D3*, *CG33639*, *Lgr4* in somatic cells, and *ETHR* and *Pdfr* signaling in the germline. In particular, CCKLR-17D3 binds DSK [59], which regulates feeding behavior and satiety [60], and somatic knockdown of *CCKLR-17D3* with four different RNAi lines led to decreased egg production compared to *Luc* RNAi control (Fig 1D). Therefore, it will be informative to test whether these preliminary findings can be reproduced using additional *UAS-hairpin RNA* lines and genetic mutants, and, if so, what steps of oogenesis are controlled by these signaling pathways.

Given the low signal-to-noise ratio in our egg count screens, it is also possible that we failed to identify some neuropeptide signaling pathways that might impact oogenesis. For instance, neuropeptides such as SIFa promote feeding and food intake [61] and, given the known effects of diet on oogenesis, disruptions in SIFa signaling pathways would be expected to affect egg production; however, our screen presumably was not sufficiently sensitive to detect these predicted effects. In addition to the roles of individual neuropeptide pathways in oogenesis, if and how neuropeptide signaling pathways crosstalk to regulate egg production should also be explored. For example, we found that $Dh31^{51}$; $Pdf^{01}$ double mutants, but not $Dh31^{51}$ or $Pdf^{01}$ single mutants, retain mature oocytes at 29°C, resulting in decreased egg laying (S5C Fig). Both *Dh31* and *Pdf* regulate circadian rhythm [62], and *Drosophila* egg laying is circadian regulated [63]. Although ablation of PDF-producing neurons does not affect egg laying circadian rhythm [64], it is possible that DH31 and PDF may act in redundant pathways to regulate circadian-regulated egg-laying behavior. Finally, although we were specifically interested in how neuropeptides produced in neurons regulate oogenesis, neuropeptides can have additional sites of production besides the nervous system [8]; therefore, it would be sensible to use a ubiquitous somatic driver such as *tub-Gal4* (instead of the neuronal-specific *nSyb-Gal4* driver) to more broadly identify neuropeptides (originating from any cell type) with potential roles in oogenesis.

## A potentially novel role for Moody/MTNR in oogenesis and regulation of stem cell number

*Drosophila moody* is required in glial cells to maintain the blood-brain barrier and mediate behavioral responses to cocaine [30,31]. Interestingly, *moody* is also required in the glia for proper male courtship behavior [65]. Nonetheless, *moody* is broadly expressed in adult *Drosophila* females [37]. Our results indicate that *moody* is not required in the niche for GSC maintenance (Fig 3I). It would be interesting to test whether *moody* is required in the glia for GSC maintenance and whether or not that involves the regulation of the blood-brain barrier. Additionally, *moody* is most strongly expressed in the spermatheca [37], and it is unknown what roles *moody* may be playing there to regulate oogenesis. There is currently no known ligand for *moody*. The closest human orthologs of *moody* are melatonin receptors 1A and 1B

(MTNR1A and MTNR1B) [29], which bind melatonin, a key regulator of circadian rhythm in vertebrates [66]. Like *moody*, melatonin receptors are expressed throughout the body, including in the central nervous system and in peripheral tissues such as the intestine, adipocytes, immune cells, epithelial tissues, ovary/granulosa cells, and myometrium [67]. In women, exogenous melatonin suppresses LH secretion and blocks ovulation via regulation of the hypothalamic-pituitary-gonadal axis, while melatonin binding to melatonin receptors on granulosa cells increases *LH* mRNA levels [67]. In culture, melatonin can also promote the proliferation of spermatogonial stem cell (SSCs) and mesenchymal stem cells (MSCs) [68,69]. In addition, given the pleiotropic role of melatonin signaling in mammals, it would be important to find out if melatonin receptors are present in SSCs and MSCs and if melatonin signaling regulates stem cell populations *in vivo*. Similarly, it would be interesting to identify the Moody ligand in *Drosophila* and determine its cellular source and requirement for GSC maintenance; to determine if Moody (and its ligand) are required for the function/behavior of other stem cells; to investigate mechanisms downstream of Moody controlling stem cells; and to pinpoint what external or physiological conditions modulate its production/secretion.

## Materials and methods

### *Drosophila* strains, culture conditions, and RNAi-based screen

Stocks were maintained at room temperature (22–25˚C) on standard medium consisting of cornmeal, molasses, yeast, and agar. Standard medium supplemented with wet yeast paste was used for all experiments, except for egg count assays (see below). S1 Table lists all mutant and transgenic *Drosophila* lines, including *Gal4* drivers, used in the study. *Dh31-R^f06589* and *Dh31-R^f05546* were backcrossed to isogenized *y w* for four generations and balanced over *CyO*. The *Dh31-R* deficiency line *w^1118; Df(2R)Exel7124/CyO* was not backcrossed because there is no visible eye marker to readily track the deletion [70]. There are two landing sites for the construct used in KK *UAS-hairpin RNA* lines. The majority of KK lines (75%) has a single insertion at the unannotated 30B site [32,33]. However, approximately 25% of the KK library has an additional insertion of the *UAS-hairpin RNA* construct at the 40D landing site, which can cause non-specific phenotypes due to ectopic expression of the Tiptop transcription factor [32,33]. We confirmed landing site occupancy of *moody^KK100674* using the previously described PCR-based method [32] (Fig 3C). *UAS-moody^GD709* is a *P* element based transgene randomly inserted on the second chromosome [71], while *UAS-moody^KK100674* and *UAS-moody^HMC06237* were site-specifically inserted on chromosome 2L and 3L, respectively [32,72]. Other genetic elements are described in FlyBase (www.flybase.org).

For pan-neuronal neuropeptide knockdown, control and experimental *nSyb>hairpin RNA* females were raised at 18˚C to minimize Gal4 expression during development [73]. Ubiquitous somatic knockdown was achieved using *tub^ts* [17], which combines *tub-Gal4* [74] with a temperature sensitive allele of the Gal4 inhibitor *tub-Gal80^ts* [75]. To screen for neuropeptide receptors required in the soma for egg production, females carrying *tub^ts* and *UAS-neuropeptide receptor hairpin RNA* or *UAS-Luciferase^JF01355* were raised at 25˚C. For germline-specific neuropeptide receptor knockdown, *MTD>neuropeptide receptor hairpin RNA* or *MTD>GFP^dsRNA.142* females were raised at 25˚C. Upon eclosion, zero- to two-day old females of all genotypes were paired with *y w* males and shifted to 29˚C, to promote *nSyb-Gal4*, *tub-Gal4*, and *MTD* activity, for various lengths of time. In the dissection-based GPCR screen and somatic *moody* knockdown experiments, females with *tub^ts* or *nos-Gal4::VP16; tub-Gal80^ts* and *UAS-hairpin RNA* were raised at 18˚C, the *Gal80^ts* permissive temperature, for inhibition of *Gal4* activity during development. [*tub-Gal80^ts*, however, has no effect over *nos-Gal4::VP16* activity [76].] Zero-to-two-day old females were collected after eclosion and paired with *y w*

males at 18˚C for two-to-three days then shifted to 29˚C, the *Gal80^ts* restrictive temperature, for zero, five, 10, or 15 days before dissection. For niche-specific *moody* knockdown, *hh>moody hairpin RNA* or *Luc^JF01355* females were raised at 25˚C. Zero-to-one-day old females were paired with *y w* males and shifted to 29˚C for zero, five, 10, or 15 days before dissection.

### Egg count assay

Five female flies were paired with five *y w* male flies per biological replicate, with three (pan-neuronal neuropeptide screen) or five (all other egg count experiments) replicates per experiment. Food was provided in the form of wet yeast paste evenly spread on molasses agar plates, and changed once (pan-neuronal neuropeptide screen) or twice (all other egg count experiments) daily. The number of eggs laid in a 24-hour period was counted on days five, 10, and 15 (RNAi), or all 15 days of the experiment (mutants), and the average number of eggs laid per female per day was calculated. Student's *t*-test (Microsoft Excel) was used to determine statistical significance.

### Ovary dissection and immunostaining

Ovaries were dissected in Grace's insect medium with L-glutamine (Caisson Labs). After teasing ovarioles apart, ovaries were fixed by nutating for 13 minutes at room temperature in 5.3% formaldehyde (Ted Pella) diluted in Grace's medium. Ovaries were rinsed three times and washed three times for 15 minutes in PBSTx (PBS; 10 mM $NaH_2PO_4$/$NaHPO_4$, 175 mM NaCl, pH 7.4, plus 0.1% Triton X-100) then blocked for three hours in blocking solution consisting of 5% normal goat serum (NGS, MP Biomedicals) and 5% bovine serum albumin (BSA, Sigma-Aldrich) in PBSTx. Samples were then incubated overnight at room temperature in mouse monoclonal anti-α-Spectrin (3A9) (DSHB; Developmental Studies Hybridoma Bank, 1:25) and mouse monoclonal anti-Lamin C (LC28.26) (DSHB, 1:25) in blocking solution. Ovaries were rinsed and washed three times in PBSTx before incubating for two hours at room temperature in Alexa Fluor 488- or 568-conjugated goat anti-mouse secondary antibodies (ThermoFisher Scientific; 1:400) in blocking solution. Following three more rinses and washes in PBSTx, ovaries were mounted in Vectashield with 1.5 µg/mL 4',6-diamidino-2-phenylindole (DAPI) (Vector Laboratories). Samples were imaged using a Zeiss LSM700 confocal microscope or Zeiss AxioImager-A2 fluorescence microscope. Whole ovary images were obtained with a Zeiss Axiocam ERc 5s camera mounted on a Zeiss Stemi 2000-CS dissecting microscope.

For EdU incorporation, ovaries were dissected in room temperature Grace's medium and incubated for one hour at room temperature in 100 µM EdU from the Click-iT EdU Alexa Fluor 594 Imaging Kit (ThermoFisher Scientific) in Grace's medium. Ovarioles were then teased apart, fixed, washed, blocked, and incubated in primary antibody as described above. Samples were subjected to the Click-iT reaction according to manufacturer's instructions, then rinsed four times and washed four times for 15 minutes before incubating in secondary antibodies, washed, and mounted in Vectashield with DAPI, as described above.

For ApopTag labeling, ovaries were dissected and fixed as described above, then washed for 30 minutes in PBSTx. ApopTag Fluorescein Direct *In Situ* Apoptosis Detection Kit (Millipore Sigma) was used according to the manufacturer's instructions. Briefly, ovaries were washed twice for five minutes in equilibration buffer, then incubated for one hour at 37˚C in TdT solution (reaction buffer plus TdT enzyme) with occasional resuspension by light tapping. Ovaries were washed in Stop/Wash buffer twice for five minutes, then rinsed and washed three times in PBSTx before mounting in Vectashield with DAPI. To quantify death of early germline

cysts, ApopTag-positive germaria were counted as a percentage of all germaria analyzed. Statistical significance of differences in the percentage of ApopTag-positive germaria across three independent experiments (100 germaria per experiment) was determined using Student's *t*-test (Microsoft Excel).

## GSC and cap cell quantification

Cap cells were identified by their ovoid shape and Lamin C-positive staining of their nuclear lamina. GSCs were identified by their direct contact with cap cells and juxtaposition of their fusome (a specialized organelle labeled by α-Spectrin [77]) to the GSC-cap cell interface, as previously described [78]. Two-way ANOVA with replication (also known as two-way ANOVA with interaction) (Microsoft Excel) was used to determine statistical significance of differences in the rate of cap cell and GSC loss over time for three independent experiments as described [20]. Student's *t*-test (Microsoft Excel) was used to determine statistical differences in the percentage of EdU-positive GSCs for each genotype for three independent experiments.

## RT-PCR and qRT-PCR

Guts (S3A Fig), heads (Fig 2C–2F and S3B Fig), or ovarectomized females (Fig 3B) were dissected in RNA*later* Stabilization Solution (ThermoFisher Scientific) and placed on ice for at least 30 minutes. Gut-derived RNA was used for experiments in S3A Fig to obtain a more robust signal as *Dh31-R* is most strongly expressed in the gut in adult flies (www.flybase.org). To extract RNA, 250 μL lysis buffer from the RNAqueous-4PCR Total RNA Isolation kit (ThermoFisher Scientific) was added to each sample, and a motorized pestle was used to homogenize tissues. RNA was purified from the homogenate following the manufacturer's instructions. cDNA was synthesized from 500 ng of total RNA for each sample using Super-Script II Reverse Transcriptase (ThermoFisher Scientific) according to the manufacturer's instructions. S2 Table lists all primers used in this study. *Rp49* primers were used as control. To quantify *Dh31* or *Dh31-R* band intensity, ImageJ was used to measure net band intensity (by subtracting background pixels from band pixels in a fixed-size box) and normalized to the corresponding *Rp49* control band. Normalized *Rp49* band intensities were set at one, and experimental sample band intensities were normalized to *Rp49*.

PowerUp SYBR Green Master Mix (ThermoFisher Scientific) was used for quantitative RT-PCR. The reactions were performed in triplicate using the QuantStudio 3 Real-Time PCR System (ThermoFisher Scientific). Examples of amplification and melt curves obtained are plotted in S8 Fig. Amplification fluorescence threshold was determined by QuantStudio 3 software, and ΔΔCt were calculated using Microsoft Excel. *Rp49* transcript levels were used as reference. Fold change of transcript levels were calculated using the equation $2^{-\Delta\Delta Ct}$ (Microsoft Excel). *y w* was used as control for analysis of mutant RNA levels; $tub^{ts} > Luc^{JF01355}$ was used as control for analysis of RNAi efficiency.

## Supporting information

**S1 Fig. Genetic background of control *UAS-hairpin RNA* lines influences egg production.** (A,B) Average number of eggs laid per female per day for females with $tub^{ts}$ (A) or *MTD-Gal4* (B) driving different control *UAS-hairpin RNA* transgenes, raised at 25˚C, and switched to 29˚C for five, 10, or 15 days. Data shown as mean±s.e.m. (TIFF)

**S2 Fig. Germline-specific RNAi-based screen for neuropeptide receptors that regulate *Drosophila* oogenesis.** *MTD* was used to drive *UAS-hairpin RNA* against neuropeptide receptor

genes, and the number of eggs laid per female per day was counted on days five, 10, and 15. *MTD>GFP^{dsRNA.142}* served as negative control. *InR* knockdown served as an internal control. *$p<0.05$; **$p<0.01$; ***$p<0.001$; ****$p<0.0001$, Student's *t*-test. Data shown as mean±s.e.m. (TIFF)

**S3 Fig. Pan-neuronal *Dh31* RNAi and ubiquitous somatic *Dh31-R* RNAi lead to variable effects on egg production that are not consistent with knockdown efficiency.** (A) Representative gel (left) and quantification (right) of RT-PCR analysis of *Dh31-R* transcript levels in female guts at seven days of ubiquitous somatic knockdown of *Dh31-R* or *Luc* control. *Rp49* was used as a control. For each genotype, the ratio of *Dh31-R* band intensity (1:1 dilution) relative to *Rp49* intensity (1:10 dilution) was normalized to that of *tub^{ts}>Luc^{JF01355}*, which was arbitrarily set at one. Three biological replicates were used for each genotype, with 10 guts per biological replicate. Gut-derived RNA was used as *Dh31-R* is most highly expressed in the gut in adults (www.flybase.org). (B) RT-PCR analysis of *Dh31* transcript levels in female heads at seven days of ubiquitous somatic knockdown of *Dh31* or *Luc* control. *Dh31* relative to *Rp49* band intensity normalized to *tub^{ts}>Luc^{JF01355}* control was calculated as described in (A). Three biological replicates were used for each genotype, with 10 heads per biological replicate. Data shown as mean±s.e.m. (C) Graph showing the average number of eggs laid per female per day at five, 10, and 15 days of ubiquitous somatic knockdown of *Dh31-R* or *Luc* control. (D) Graph showing the average number of eggs laid per female per day at five, 10, and 15 days of pan-neuronal RNAi of *Dh31* or *Luc* control. *$p<0.05$; **$p<0.01$; ***$p<0.001$; ****$p<0.0001$, Student's *t*-test. Data shown as mean±s.e.m. (TIFF)

**S4 Fig. *Dh31* and *Dh31-R* mutants have similar rates of egg production as heterozygous controls.** (A,B) Line graphs showing the average number of eggs laid per female per day at different days after eclosion for *Dh31-R* mutants and heterozygous controls at 25˚C (A) or 29˚C (B). Data from days five, 10, and 15 at 25˚C are also shown in Fig 2G. Note that in (B), the two homozygous mutants and transheterozygous mutant lay fewer eggs at 29˚C; this is likely due to linked background mutation as neither *Dh31-R^{f06589}/Df(2R)Exel7124* nor *Dh31-R^{f05546}/Df (2R)Exel7124* lay fewer eggs than heterozygous controls. (C,D) Line graphs showing the average number of eggs laid per female per day for *Dh31* mutants and heterozygous controls at 25˚C (C) or 29˚C (D). Data from day five, 10, and 15 at 25˚C are also shown in Fig 2H. Data shown as mean±s.e.m. (TIFF)

**S5 Fig. *Dh31^{51}; Pdf^{01}* double mutants lay fewer eggs at 29˚C due to mature oocyte retention in their ovaries.** (A,B) Line graphs showing the average number of eggs laid per female per day at different days after eclosion for *Dh31^{51}* homozygous, *Pdf^{01}* homozygous, or *Dh31^{51}; Pdf^{01}* double homozygous females at 25˚C (A) or 29˚C (B). Data shown as mean±s.e.m. (C) Examples of ovaries from *Dh31^{51}*, *Pdf^{01}*, and *Dh31^{51}; Pdf^{01}* 5-day-old females at 25˚C (left) or 29˚C (right). Arrows point to multiple dorsal appendages to indicate examples of accumulated mature oocytes. (TIFF)

**S6 Fig. *Dh31*, *Dh31-R*, and *Pdf* mutants do not show increased levels of early germline cyst death relative to control females.** Percent of germaria showing ApopTag-positive germline cysts in five-day old *Dh31-R* mutant (A), *Dh31* mutant (B), or *Dh31^{51}; Pdf^{01}* females at 25˚C (C) or 29˚C (D). Three biological replicates per genotype, 100 germaria per replicate. **$p<0.01$, Student's *t*-test. Data shown as mean±s.e.m. (TIFF)

**S7 Fig. Somatic *moody* and germline *AstC-R1* do not appear to regulate GSC proliferation.** (A) Example of germarium from *MTD>GFP^dsRNA.142^* female at 7 days of *GFP* knockdown showing one EdU-positive GSC (arrowhead) and one EdU-negative GSC. DAPI (blue) labels nuclei. LamC (green), nuclear lamina of cap cells; α-Spectrin (green), fusome; EdU (red), S-phase marker. Cap cells, arrows; GSCs, solid outlines. Scale bar, 10 μm. (B,C) Graphs showing the average percentage of EdU-positive GSCs at zero or 10 days of somatic knockdown of *moody* or *Luc* control (B) or at seven days of germline knockdown of *AstC-R1* or *GFP* (C). Two biological replicates for *moody^HMC06237^*, *GFP^dsRNA.142^*, and all *Astc-R1* RNAi genotypes, and three biological replicates for all other genotypes. The total number of GSCs analyzed are indicated inside bars.
(TIFF)

**S8 Fig. Examples of qPCR amplification and melt curves.** (A) Amplification curves for three biological replicates of *Dh31-R^f05546^/+*, with three technical replicates per biological sample, plotting Rn versus cycle number, with amplification for *Dh31-R* shown in shades of blue and *Rp49* in shades of gray. (B) Melt curves for the same sample of *Dh31-R^f05546^/+* plotting the derivative of fluorescence intensity versus temperature, with *Dh31-R* products in shades of blue and *Rp49* products in shades of gray. Each line represents one technical replicate of a biological replicate. Biological replicates were collected at different times, and qPCR was performed separately.
(TIFF)

**S1 Table. Transgenic *Drosophila* lines used in this study.**
(PDF)

**S2 Table. Sequences of primers used in this study.**
(PDF)

**S1 File. Results from egg count screens.**
(XLSX)

## Acknowledgments

T.M. and D.D.-B. designed experiments, analyzed and interpreted data, and wrote the manuscript. The initial dissection-based screen was performed by S.M.; T.M. performed subsequent analyses of *moody* and all other experiments. We thank Metabel Markwei for her technical assistance with egg count experiments shown in S3 Fig. We are grateful to the Developmental Studies Hybridoma Bank for antibodies (National Institutes of Health [NIH] N01HD073263-000), and to the Bloomington Stock Center (NIH P400D018537), Vienna *Drosophila* Stock Center, Mark Wu, Erika Matunis, and Fumika Hamada for *Drosophila* stocks. We thank Lesley N. Weaver and Rodrigo Dutra Nunes for critical reading of the manuscript.

## Author Contributions

**Conceptualization:** Tianlu Ma, Daniela Drummond-Barbosa.

**Formal analysis:** Tianlu Ma, Shinya Matsuoka.

**Funding acquisition:** Daniela Drummond-Barbosa.

**Investigation:** Tianlu Ma.

**Methodology:** Tianlu Ma.

**Project administration:** Daniela Drummond-Barbosa.

**Supervision:** Daniela Drummond-Barbosa.

**Validation:** Tianlu Ma.

**Visualization:** Tianlu Ma.

**Writing – original draft:** Tianlu Ma.

**Writing – review & editing:** Tianlu Ma, Daniela Drummond-Barbosa.

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
