## [Decision Letter · Decision Letter 0]

13 Nov 2020

PONE-D-20-32229

RNAi-based screens uncover a potential new role for the orphan neuropeptide receptor Moody in Drosophila female germline stem cell maintenance

PLOS ONE

Dear Dr. DRUMMOND-BARBOSA,

Thank you for submitting your manuscript to PLOS ONE. After careful consideration, we feel that it has merit but does not fully meet PLOS ONE’s publication criteria as it currently stands. Therefore, we invite you to submit a revised version of the manuscript that addresses the points raised during the review process.

As you will see from the comments, both reviewers find your study technically sound and well conducted, and your conclusions fully backed up by the data. Thus, the major publication criteria for PLOS ONE are basically met. One of the reviewers is nevertheless suggesting sensible additional experiments which will make the story stronger. On the other had, you may see this manuscript reporting an RNAi-screen as a platform for a larger independent and more in-depth study. As publication criteria for PLOS ONE do not include "impact", we will therefore not ask for these extra experiments but of course they are welcome if you want to include them.

We look forward to receiving your revised manuscript.

Kind regards,

Christian Wegener

Academic Editor

PLOS ONE

Journal Requirements:

Reviewers' comments:

Reviewer's Responses to Questions

**Comments to the Author**

1. Is the manuscript technically sound, and do the data support the conclusions?

Reviewer #1: Yes

Reviewer #2: Yes

2. Has the statistical analysis been performed appropriately and rigorously? 

Reviewer #1: Yes

Reviewer #2: Yes

3. Have the authors made all data underlying the findings in their manuscript fully available?

Reviewer #1: Yes

Reviewer #2: Yes

4. Is the manuscript presented in an intelligible fashion and written in standard English?

Reviewer #1: Yes

Reviewer #2: Yes

5. Review Comments to the Author

Reviewer #1: In their work, the authors tested neuropeptides and (putative) neuropeptide receptors for their function in oogenesis. They expressed RNAi construct in neurons, somatic cells and germ cells and determined the number of eggs laid. They identify potential candidates and follow up on Dh31 and the corresponding receptor, Dh31-R, and on moody.

Dh31 and Dh31-R RNAis showed effects in neuronal (with nsyb-Gal4) and somatic knockdown (with tubulin-Gal4; Tubulin-Gal80ts which the authors named tubts). When the authors followed up with mutants, these findings could not consistently be confirmed, and the severity of the mutants (in terms of RNA levels) did not correlate with observed effects. A Dh3151 / Pdf01 double mutant may affect egg retention. Overall, the results were not consistent and the author concluded that the two genes probably play no role in oogenesis.

Somatic Moody RNAi expression resulted in reduced egg numbers. Dissected ovaries showed reduced numbers of GSCs, but no reduction in cap cells. These findings suggest a role for moody in oogenesis. But since a repeat experiment examining more dissected ovaries did not consistently confirm these findings, this role is uncertain.

So while the authors find potential candidates, they conclude that screening for genes affecting oogenesis using egg-laying is difficult since the signal-to noise ratio is low. The authors suggest that genetic background may play a big role, as well as a network with many players so that the effect of one mutant player may not reliably show up in the number of eggs laid. A more sensitive approach, yet more labor intensive, may be to examine dissected ovaries.

The study was performed with great care to detail and a lot of data were generated. Several RNAi lines were tested, as well as mutants, and for many of them RNA levels were determined. The conclusions are generally well supported by the results. A few points could use some clarification (see detailed suggestions below). It is valuable to show the pitfalls of this approach in studying oogenesis, especially since the studies are done very carefully. Potential reasons and conclusions are well described.

The text of the manuscript could be more streamlined. For example, the importance of neuropeptides, and the connection to mammalian systems, is mentioned in detail repeatedly.

In contrast, a few places could use additional clarification.

Specific suggestions:

1. Fig. 2I: A description of what ApopTag labels would be useful in the figure legend. An arrow pointing out differences would be useful. I fail to see a difference between the control and the mutant. A better example for the observed difference is needed.

2. Please describe what the “maternal triple driver MTD” is.

3. For Dh3151 /Pdf01 double mutants the authors say that the reduced number of eggs laid might be due to egg retention. As evidence, they show the pictures of dissected ovaries and say that egg appendices can be seen in the mutants. This is not visible in the picture and a magnification of these ovaries should be shown, with arrows pointing to these structures.

4. For moody, the authors say that in a separate screen they found reduced numbers of GSCs. This result is listed in the included table, but no data are shown. Furthermore, the authors say that in a later experiment with larger sample size, this could not be repeated. Since data are not shown, and could not be repeated, these findings should not be discussed as supporting evidence, but as a caveat for the RNAi findings.

5. The authors mention that it will be interesting to study the downstream pathway of moody signaling. In development, there is evidence that this involves cAMP and PKA signaling. Is it known whether these pathways are needed for oogenesis, and in which cells?

Reviewer #2: Summary and Critique

Reproductive physiology is maintained by a variety of nutritional and neural signals. However, regulation of these complex endocrine pathways has not been completely described. In this study, Ma and Drummond-Barbosa use RNAi-based approaches to screen for novel neural regulators of reproductive output (egg production). Unfortunately, their screens only identified a few potential regulators. However, the data collection is solid, the writing is top-notch, and publication of the data may form the foundation for other, more targeted studies in the future. Importantly, Ma and Drummond-Barbosa show that the neuropeptide receptor Moody is necessary in the soma for germline stem cell number, although the mechanisms by which this occurs remain to be elucidated. Given that this is the report of a genetic screen, the descriptive nature of the study is appropriate. The experiments are technically rigorous and the results described sufficiently. Overall, the text is well-written, and the conclusions are supported by the data presented. I have only minor comments for revision, listed below in order of appearance.

1. Results, lines 196-199. It is worth mentioning in the results/discussion of Dh31 and Pdf single and double mutants that the defects in egg laying or egg retention may suggest roles in ovulation, vs roles in egg production per se.

2. Results, lines 282-285. The authors bring up insulin signaling to cap cells as a possible connection for why Moody mutants have fewer GSCs. This seems like a stretch to me. It is perhaps more reasonable to think that Moody, as a GPCR, is expressed in ovarian somatic cells (cap cells or otherwise) and thus regulates GSC number…maybe not affecting cap cell number, but rather cap cell interconnectivity or interaction with GSCs. It looks like a UAS-moody-RFP and a moody-Gal4 have been published elsewhere (Schwabe et al., Cell, 2005, 123(1):133-44). If possible, the authors should test whether Moody is expressed in ovarian somatic cells.

3. Since most of the paper describes the RNAi screens, I am hesitant to suggest additional experiments. However, if the authors wanted to pursue Moody to more deeply understand why it is necessary in somatic cells to maintain GSC numbers, it would be interesting to use additional somatic drivers in the ovary and ovarian muscle sheath to test whether Moody impacts actin-dependent septate junction maintenance (either in cap cells or escort cells). This has been demonstrated for Moody in the maintenance of the blood brain barrier, and similar septate junction proteins (like Coracle, for example) are expressed in escort cells. A stronger somatic driver (like c587-Gal4) or one more specific to anterior escort cells (like Pdk1-Gal4) might also better recapitulate the tubulin-Gal4 finding than hh-Gal4, which is relatively weak.

4. Figure 3, panel I: there is a small asterisk at the 15d timepoint, but it is unclear to me whether this is indicating statistical significance? The timepoints seem to overlap?

6. PLOS authors have the option to publish the peer review history of their article (what does this mean?). If published, this will include your full peer review and any attached files.

Reviewer #1: No

Reviewer #2: No

---

## [Author Response · Author response to Decision Letter 0]

20 Nov 2020

PONE-D-20-32229 - RESPONSE TO REVIEWERS

REVIEWER 1

Reviewer 1 acknowledged: “The study was performed with great care to detail and a lot of data were generated. Several RNAi lines were tested, as well as mutants, and for many of them RNA levels were determined. The conclusions are generally well supported by the results. (…) It is valuable to show the pitfalls of this approach in studying oogenesis, especially since the studies are done very carefully. Potential reasons and conclusions are well described.”

He/She also said that a few points should be clarified, as detailed below. 

Point 1: “Fig. 2I: A description of what ApopTag labels would be useful in the figure legend. An arrow pointing out differences would be useful. I fail to see a difference between the control and the mutant. A better example for the observed difference is needed.”

We added a sentence in the figure legend explaining that dying cysts are labeled by ApopTag and found in both control and Dh31-R knockdown germaria, and these dying cysts look similar based on ApopTag staining (lines 178-179 of “clean” manuscript or lines 195-196 of manuscript with track changes). 

Point 2: “Please describe what the “maternal triple drive MTD is.”

We have included information about the three Gal4 drivers that make up MTD (lines 109-110 or “clean” manuscript or lines 120-121 of manuscript with track changes).

Point 3: “For Dh3151 /Pdf01 double mutants the authors say that the reduced number of eggs laid might be due to egg retention. As evidence, they show the pictures of dissected ovaries and say that egg appendices can be seen in the mutants. This is not visible in the picture and a magnification of these ovaries should be shown, with arrows pointing to these structures.”

We have added more arrows to S5C Fig and made sure they point directly at the dorsal appendages of multiple examples of accumulated mature oocytes to help readers visualize these structures. 

Point 4: “For moody, the authors say that in a separate screen they found reduced numbers of GSCs. This result is listed in the included table, but no data are shown. Furthermore, the authors say that in a later experiment with larger sample size, this could not be repeated. Since data are not shown, and could not be repeated, these findings should not be discussed as supporting evidence, but as a caveat for the RNAi findings.”

We initially identified moody in a separate screen, where knockdown of moody appeared to both reduce the number of GSCs and increase the percentage of EdU-positive GSCs. Later experimentation with larger sample sizes showed that while the reduction in GSC number was reproducible (Fig 3D-F), we no longer saw any statistically significant increase in the percentage of EdU-positive GSCs (S7 Fig). Results from the screen are described in lines 251-253 (of manuscript with track changes), and follow-up experiments for EdU incorporation are described in lines 253-255 (of manuscript with track changes). We have added “EdU incorporation” in line 254 (of manuscript with track changes) to clarify the nature of these experiments, and implications of these findings (the requirement of larger sample sizes to accurately capture changes in GSC proliferation) are discussed in lines 397-404 (of manuscript with track changes).

Point 5: “The authors mention that it will be interesting to study the downstream pathway of moody signaling. In development, there is evidence that this involves cAMP and PKA signaling. Is it known whether these pathways are needed for oogenesis, and in which cells?”

PKA and cAMP signaling has been shown to be required in oocytes for microtubule cytoskeleton to reorganize properly during mid-oogenesis (Steinhauer and Kalderon, Development, 2005). However, there is no evidence in the literature that PKA signaling regulates GSC number. 

REVIEWER 2

Reviewer 2 stated: “In this study, Ma and Drummond-Barbosa use RNAi-based approaches to screen for novel neural regulators of reproductive output (egg production). Unfortunately, their screens only identified a few potential regulators. However, the data collection is solid, the writing is top-notch, and publication of the data may form the foundation for other, more targeted studies in the future. Importantly, Ma and Drummond-Barbosa show that the neuropeptide receptor Moody is necessary in the soma for germline stem cell number, although the mechanisms by which this occurs remain to be elucidated. Given that this is the report of a genetic screen, the descriptive nature of the study is appropriate. The experiments are technically rigorous and the results described sufficiently. Overall, the text is well-written, and the conclusions are supported by the data presented.” 

He/She had minor comments for revision, addressed below.

Point 1: “Results, lines 196-199. It is worth mentioning in the results/discussion of Dh31 and Pdf single and double mutants that the defects in egg laying or egg retention may suggest roles in ovulation, vs roles in egg production per se.” 

We have added a sentence in lines 225 (of manuscript with track changes) potentially suggesting a role for Dh31 and Pdf in ovulation.

Point 2: “Results, lines 282-285. The authors bring up insulin signaling to cap cells as a possible connection for why Moody mutants have fewer GSCs. This seems like a stretch to me. It is perhaps more reasonable to think that Moody, as a GPCR, is expressed in ovarian somatic cells (cap cells or otherwise) and thus regulates GSC number…maybe not affecting cap cell number, but rather cap cell interconnectivity or interaction with GSCs. It looks like a UAS-moody-RFP and a moody-Gal4 have been published elsewhere (Schwabe et al., Cell, 2005, 123(1):133-44). If possible, the authors should test whether Moody is expressed in ovarian somatic cells.”

Insulin signaling, which has a well-known role in regulating GSC number, is dependent on signals such as Unpaired 2 (Upd2) traveling past the blood-brain barrier to insulin-producing cells (IPCs) in the brain. Since Moody plays an essential role in maintaining the blood-brain barrier, we feel that it was reasonable to rule out if the GSC loss phenotype was due to global insulin signaling being disrupted due to moody knockdown. Regardless, our results indicated that GSC loss was not due to changes in insulin signaling, and we explored whether moody is required in somatic cells in the germarium using hh-Gal4 (and found that it is not). (See lines 327-332 of manuscript with track changes.)

Point 3: “Since most of the paper describes the RNAi screens, I am hesitant to suggest additional experiments. However, if the authors wanted to pursue Moody to more deeply understand why it is necessary in somatic cells to maintain GSC numbers, it would be interesting to use additional somatic drivers in the ovary and ovarian muscle sheath to test whether Moody impacts actin-dependent septate junction maintenance (either in cap cells or escort cells). This has been demonstrated for Moody in the maintenance of the blood brain barrier, and similar septate junction proteins (like Coracle, for example) are expressed in escort cells. A stronger somatic driver (like c587-Gal4) or one more specific to anterior escort cells (like Pdk1-Gal4) might also better recapitulate the tubulin-Gal4 finding than hh-Gal4, which is relatively weak.”

We agree that it would be interesting in future studies to pursue moody using additional somatic drivers, both in the ovary and in other tissues to determine where moody is required for GSC maintenance and the mechanism through which moody regulates GSC numbers (whether through regulating septate junction proteins or other downstream pathways). 

Point 4: “Figure 3, panel I: there is a small asterisk at the 15d timepoint, but it is unclear to me whether this is indicating statistical significance? The timepoints seem to overlap?”

There is an asterisk in Panel I for moodyKK100674, indicating statistical significance as determined by two-way ANOVA with replication. This method measures how two independent variables (in this case, genotype and time) interact to affect the dependent variable (the number of GSCs) and takes into account the slope of the curve. The statistical significance likely reflects the difference in slopes between moodyKK100674 and LucJF01355 from days 5 to 10 and from 10 to 15, though, as the reviewer points out, the overall slope of the two lines (from days 0 to 15) are very similar, which supports our conclusion that moody is likely not required in the niche for GSC maintenance.

ACADEMIC EDITOR/JOURNAL REQUIREMENTS:

“Please ensure that your manuscript meets PLOS ONE's style requirements, including those for file naming.”

We have updated our manuscript formatting and file names to meet PLOS ONE’s style requirements. 

“While revising your submission, please upload your figure files to the Preflight Analysis and Conversion Engine (PACE) digital diagnostic tool, https://pacev2.apexcovantage.com/. PACE helps ensure that figures meet PLOS requirements.”

We have done so.

---

## [Editor Report · Decision Letter 1]

26 Nov 2020

RNAi-based screens uncover a potential new role for the orphan neuropeptide receptor Moodyin Drosophila female germline stem cell maintenance

PONE-D-20-32229R1

Dear Dr. DRUMMOND-BARBOSA,

We’re pleased to inform you that your revised manuscript has been judged scientifically suitable for publication and will be formally accepted for publication once it meets all outstanding technical requirements.

Thank you very much for publishing your results with PLOS ONE. Personally, I hope you continue your line of research on neuropeptides/neuropeptide receptor function in oogenesis, based on the results and experiences of the reported RNAi screen.

Kind regards,

Christian Wegener

Academic Editor

PLOS ONE

---

## [Editor Report · Acceptance letter]

3 Dec 2020

PONE-D-20-32229R1 

RNAi-based screens uncover a potential new role for the orphan neuropeptide receptor Moody in *Drosophila* female germline stem cell maintenance 

Dear Dr. Drummond-Barbosa:

I'm pleased to inform you that your manuscript has been deemed suitable for publication in PLOS ONE. Congratulations! Your manuscript is now with our production department. 

Kind regards, 

on behalf of

Prof. Dr. Christian Wegener 

Academic Editor

PLOS ONE